# Motion Analysis Focusing on Rotational Movements of Professional Female Baseball Pitchers: Comparison with Male University Baseball Pitchers

**DOI:** 10.3390/ijerph182413342

**Published:** 2021-12-18

**Authors:** Yoshikazu Azuma, Tomoyuki Matsui, Machiko Hiramoto, Ruo Hashimoto, Kanta Matsuzawa, Tetsuya Miyazaki, Kazuya Seo, Yuya Watanabe, Noriyuki Kida, Yoshihiro Kai, Toru Morihara

**Affiliations:** 1Marutamachi Rehabilitation Clinic, Kyoto 604-8405, Japan; matsui.tomoyuki.sports.reha@gmail.com (T.M.); true.to.your.heart810@gmail.com (M.H.); ruo.hashimoto@gmail.com (R.H.); k.matsuzawa.pt@gmail.com (K.M.); mtsports0512@gmail.com (T.M.); toru4271@koto.kpu-m.ac.jp (T.M.); 2Rehabilitation Unit, University Hospital, Kyoto Prefectural University of Medicine, Kyoto 602-8566, Japan; seo@koto.kpu-m.ac.jp; 3Faculty of Health and Sports Science, Doshisha University, Kyotanabe 610-0394, Japan; yuwatana@mail.doshisha.ac.jp; 4Department of Biotechnology, Graduate School of Science and Technology, Kyoto Institute of Technology, Kyoto 606-8585, Japan; kida@kit.ac.jp; 5Department of Physical Therapy, Faculty of Health Sciences, Kyoto Tachibana University, Kyoto 607-8175, Japan; kai-y@tachibana-u.ac.jp

**Keywords:** kinematic data, pelvis, thorax, lateral tilt

## Abstract

Purpose: The purpose of this study was to compare pitching motion of the professional female baseball pitchers with the male university baseball pitchers focused on the pelvic and thoracic movements. Subjects and methods: The participants were 15 healthy professional female baseball pitchers (11 right-handers and 4 left-handers; age, 21.7 ± 3.2 years; height, 162.5 ± 5.1 cm; weight, 59.0 ± 6.6 kg) and 14 healthy male university baseball pitchers (12 right-handers and 2 left-handers; age, 19.9 ± 0.8 years; height, 176.4 ± 3.0 cm; body mass, 73.1 ± 3.0 kg). Throwing motion was captured by three-dimensional motion analysis system. Kinematic data of the lead hip, pelvis, thorax, and dominant shoulder were collected and the joint angle at maximum external rotation phase and ball release phase were compared. Results: The female baseball pitchers rotated pelvis and thorax more than the male at the maximum external rotation phase and ball release phase (*p* < 0.05). At the same, the pelvis and thorax of the female baseball pitchers were tilted significantly closer to horizontal plane than the male (*p* < 0.05). The pelvis and thorax of the male baseball pitchers was tilted to non-dominant lateral side. Conclusions: The results of this study indicate that the pelvic and thoracic movements of the professional female baseball pitchers was different from male university pitchers.

## 1. Introduction

Women’s baseball is popular in Japan. The Japan Women’s Baseball League was established in 2010. Moreover, the number of high school girls’ baseball clubs have increased from five in 2007 to 24 in 2017 [1]. Moreover, the number of female baseball players is steadily increasing. Furthermore, the performance of female Japanese baseball players is high, taking the first place on six occasions and the second place twice in the Women’s Baseball World Cup, which has been held since 2004. Investigating the characteristics of high-performing players might contribute to improving the performance of female baseball players and the prevention of pitching-related injuries.

Many studies examined the throwing motion. The throwing motion is performed by using every part of the body [2]. In these motions, energy is transmitted from the trunk to the upper extremity between the foot contact of the lead leg (FC) and maximum external rotation of the shoulder joint in the throwing-side (MER), and from the upper extremity to ball between MER and ball release (BR) [2]. Different variables have been used in the evaluation of throwing motions including step length when throwing the ball, the position of the FC, orientation of the pelvis and thorax, anterior and posterior as well as lateral trunk tilt, angles of the shoulder and elbow joints, the angular velocity, the timing of the maximum angle, and the time between FC and BR, among others [3].

Among the variable listed above, movements on the horizontal plane are important. The speed of the ball is correlated with the orientation of the pelvis and thorax at MER, and with pelvic orientation at BR [4]. The range of trunk rotation of baseball players that have experienced pitching-related injuries is narrower than those without such injuries [5]. Moreover, a decline of the range of rotation in the throwing direction of the neck, trunk, and the range of hip internal rotation of the lead leg might become a risk factor for pitching-related injuries [6].

Movements of the pelvis and thorax should be evaluated independently in the evaluation of trunk movements. However, previous studies on the analysis of throwing motion have independently evaluated only the movements on the horizontal plane of the pelvis and thorax. Moreover, the pelvis and thorax have not been independently evaluated in the evaluation of anterior and posterior as well as lateral tilt. In addition, previous studies have focused on male baseball pitchers.

The purpose of this study was to compare pitching motion of the professional female baseball pitchers with the male university baseball pitchers focused on the pelvic and thoracic movements.

## 2. Material and Methods

### 2.1. Participants

Professional female baseball pitchers (N = 18) that could throw a ball with their full strength (N = 15) participated in this study (the female player group). They were compared with male university baseball pitchers (the male player group, N = 14). See Table 1 for the attributes of the participants. Inclusion criteria included no history of shoulder or elbow pain that involved time loss from competition in the previous one year. Sidearm pitchers and submarine pitchers were excluded from the analysis and the data of overhand or three-quarter pitchers were analyzed.

### 2.2. Methods

A three-dimensional motion analysis device (VICON MX^®^, Vicon Motion Systems Ltd., Oxford, England) was used for measurement. It consisted of nine infrared cameras set-up in a laboratory. The sampling frequency was 250 Hz [7]. Thirty-nine infrared light reflecting markers were pasted on the body of each participant following the plug-in-gait model [8]. After their preferred sufficient preferred warm-up period (e.g., static and dynamic stretching, throwing exercises, and pitching specific exercises), the participants stood barefooted on level ground, threw a ball towards the net that was set 5 m ahead with their full strength. The speed of the ball was measured using a speed gun (Stalker Sports II, Applied Concepts Ltd., Plano, TX, USA) placed behind the net. The measurement was conducted five times, and the data of the fastest trial were analyzed. The analysis was conducted at MER and BR [9,10].

### 2.3. Parameters

The following parameters were examined; angles of the throwing-side shoulder joint (abduction, external rotation; Figure 1), angles of the orientation of the pelvis and thorax (Figure 2), angles of the tilt of the pelvis and thorax (anterior and posterior and lateral; Figure 2), and angles of the hip joint of the lead leg (flexion, adduction, and internal rotation; Figure 3).

### 2.4. Definition

Alignments on the three-dimensional space coordinates were calculated for the pelvis and thorax angles according to a define described previously [11,12]. The pitching direction (the direction towards the catcher) was regarded as the longitudinal axis of the coordinates, the line between the first base and the third base as the lateral axis, and the vertical line to the ground as the vertical axis. The mid-point between right and left anterior superior iliac spines of the pelvis was regarded as the center of the axis. Moreover, the line connecting the mid-point between right and left posterior superior iliac spines and the mid-point between right and left anterior superior iliac spines were considered as the longitudinal axis. Furthermore, the line connecting right and left anterior superior iliac spines as the lateral axis, and the vertical line to the plane consisting of the lateral and longitudinal axes as the vertical axis. In addition, in for thorax, the line connecting the mid-point between C7 and T10 and the mid-point between the clavicular notch and the xiphoid process was regarded as the longitudinal axis, the line connecting the mid-point between C7 and the clavicular notch and the mid-point between T10 and the xiphoid process as the vertical axis, and the vertical line to the plane consisting of the longitudinal and vertical axes as the lateral axis. The center of the axes in the thorax was regarded as the point 7 mm behind the radius of the infrared light reflective markers that were pasted along the longitudinal axis starting from the clavicular notch. The center of the axes in the thorax was regarded as the clavicular notch.

### 2.5. Statistical Analysis

Differences in the mean values of the angles in each joint at MER and BR between the female and male pitchers were also examined. Moreover, pelvic orientation that exceeded 90 degrees was examined and compared between the two groups. When the angle was 90 degrees, the pelvis is considered to face the catcher. Statistical analysis was conducted using unpaired *t*-tests. Furthermore, Pearson’s correlation coefficient was calculated to examine the correlation between the hip joint angle and orientation, as well as the tilt of the pelvis. The significant *p*-value was regarded as 5%. All statistical procedures were calculated using the statistical package SPSS (version 175 20) (IBM, Armonk, NY, USA). The study was conducted after obtaining the approval of the ethics committee of the Kyoto Prefectural University of Medicine (RBMR-C1197-2). All experiments were conducted in accordance with the ethical standards of the Kyoto Prefectural University of Medicine and with the 1964 Helsinki Declaration and its later amendments [13].

## 3. Results

The speed of the ball in the female group was 103.0 ± 6.2 km/h, whereas that of the male group was 122.4 ± 8.3 km/h, the latter was significantly faster (*p* < 0.01). Table 2 shows the angles of the shoulder joint. The angles of abduction at MER and BR were between 90 and 100 degrees. The angles of external rotation were 160–166 degrees at MER, whereas it was approximately 130 degrees at BR. Significant differences were not indicated at both MER and BR. Female pitchers had lower values than male pitchers.

Table 3 shows the orientation of the pelvis and thorax. The pelvic orientation of female pitchers exceeded 90 degrees at MER, whereas it was below 90 degrees in male pitchers. The angle of the thorax orientation was approximately 9 degrees higher than that of the pelvis at MER and approximately 14 degrees higher at BR in both groups. The angles of the orientation of the pelvis and thorax in female pitchers were significantly higher than male pitchers by approximately 9 degrees at both MER and BR (MER, pelvis: t (27) = −4.44, *p* < 0.01, thorax: t (27) = −3.83, *p* < 0.01; BR, pelvis: t (27) = −3.41, *p* < 0.01, thorax: t (27) = −3.72, *p* < 0.01).

Table 4 shows the tilting angles of the pelvis and thorax. Regarding the anterior and posterior pelvic tilt, both groups indicated approximately 35 degrees anterior pelvic tilt at both MER and BR, which indicated no significant differences. As for the anterior and posterior tilts of the thorax, both groups indicated lower values than the pelvis by 13–18 degrees at MER, and by 5–10 degrees at BR. Regarding the thorax anterior tilt, female pitchers indicated significantly lower values than male pitchers by 5–10 degrees at both MER and BR (MER, thorax: t (27) = 2.54, *p* < 0.05; BR, thorax: t (27) = 2.50, *p* < 0.05). 

As for lateral pelvic tilt, female pitchers indicated approximately 1 degree left tilt and male pitchers indicated 9 degrees left tilt at both MER and BR. Female pitchers indicated significantly lower values (MER, t (27) = 3.68, *p* < 0.01; BR, t (27) = 3.78, *p* < 0.01). Regarding thorax lateral tilt, both groups indicated higher left tilt values than the pelvis by 20–25 degrees at both MER and BR. Female pitchers indicated significantly lower values at MER (t (27) = 2.48, *p* < 0.01). At BR, though significant differences were not shown, female pitchers indicated lower values than male pitchers (t (27) = 1.76, *p* = 0.09).

Table 5 shows the angles of the hip joint. There were no significant differences in hip joint flexion angles at both MER and BR. Regarding hip joint adduction angles, female pitchers indicated higher values by approximately 10 degrees at both MER and BR (MER, t (27) = −5.23, *p* < 0.01; BR, t (27) = −4.94, *p* < 0.01). As for hip joint internal rotation angles, female pitchers indicated lower values at both MER and BR by approximately 15 degrees (MER, t (27) = 3.19, *p* < 0.01; BR, t (27) = 3.00, *p* < 0.01) (Figure 4 and Figure 5).

Next, correlations between hip joint angles and pelvic orientation and tilt angles in both groups were examined. In female pitchers, significant positive correlations were indicated between hip joint adduction and pelvic orientation angles (r = 0.66, *p* < 0.01) as well as between hip joint flexion and anterior pelvic tilt (r = 0.82, *p* < 0.01) at MER. At BR, positive correlations were indicated between hip joint adduction and pelvic orientation angles (r = 0.73, *p* < 0.01), between hip joint flexion and anterior pelvic tilt (r = 0.70, *p* < 0.01), as well as between hip joint internal rotation and pelvic left tilt (r = 0.60, *p* < 0.05). In male pitchers, significant positive correlations were indicated between hip joint flexion and anterior pelvic tilt (r = 0.91, *p* < 0.01) as well as between hip joint flexion and pelvic left tilt (r = 0.54, *p* < 0.05) at MER. At BR. In general, hip joint adduction was highly correlated with the pelvic orientation, hip joint flexion was very highly correlated with the anterior pelvic tilt, and hip joint internal rotation was highly correlated with the left pelvic tilt.

## 4. Discussion

Characteristics of the pitching motion of professional female baseball pitchers were examined and compared with male university baseball pitchers. This study indicated that the pelvic and thoracic movements of the professional female baseball pitchers was different from male university pitchers. To the best of our knowledge, no previous study has examined throwing motion by separating pelvis from thorax. These results provide insight into the kinematic and kinetic analysis of the throwing motion focused on pelvic and thoracic movements.

### 4.1. External Rotation Angle of the Shoulder

The results of this study did not indicate significant differences in the shoulder joint angle. Previous studies have reported that the maximum external rotation angle of the shoulder joint of male university or professional baseball players were 169.0–182.6 degrees, which were identical to the results of this study [10,14,15]. Fleisig et al. [10] compared throwing motions of male elementary school students, junior/senior high school students, university students, and professional baseball players and reported that there were no significant differences in the maximum external rotation angle of the shoulder joint [10]. It is considered that the shoulder joint angle has no correlation with age or with gender.

### 4.2. Pelvic and Thorax Orientation Angle

The results of this study indicated that the angles of the orientation of the pelvis and thorax in female pitchers were significantly higher than male pitchers at both MER and BR. Stodden et al. [4] examined the orientation of the pelvis and thorax of male high school pitchers, university pitchers, and professional pitchers and reported the pelvic orientation angle was 85 degrees at MER and 89 degrees at BR, whereas the thorax orientation angle was 88 degrees at MER and 111 degrees at BR. The results of the current study on pelvic angle of male university pitchers were identical to Stodden et al. [4] which is indicative of the validity of these results. Ito et al. [16] reported that the values of pelvis orientation angles and thorax of female university baseball players were lower than male university baseball players [16]. Moreover, they were directed to the third base. However, in the present study, the values of the pelvis and thorax orientation angles of professional female baseball players were higher than male university baseball players. Moreover, they were directed to the first base. It might be possible that professional female baseball players have high pitching skills and can rotate the pelvis and thorax using their lower body and trunk. Furthermore, there were no differences in the shoulder joint angle between female and male pitchers, whereas the pelvis orientation angle of female pitchers exceeded 90 degrees at MER. It is considered that female baseball pitchers might try to effectively transform the energy of the translational motion into rotational motion for achieving high-level performance. However, excessive rotation of the pelvis and thorax before BR might lead to a pitching form dependent on the upper body. It is suggested that in the future, appropriate orientation of the pelvis and thorax should be examined in more detail.

### 4.3. Anterior or Lateral Tilt Angle of the Pelvis and Thorax

Regarding the anterior tilt of the pelvis and thorax, female pitchers’ thorax anterior tilt indicated significantly lower values than male pitchers, whereas significant differences were not shown in the pelvic anterior tilt. The pelvic anterior tilt angle was approximately 35 degrees in both groups at both MER and BR, and there were no differences between MER and BR. Moreover, the lateral tilt of the pelvis and thorax of female pitchers was lower at both MER and BR. Furthermore, the hip joint adduction was highly correlated with the pelvic orientation, and hip joint flexion was very highly correlated with the pelvis anterior tilt, and hip joint internal rotation was highly correlated with pelvis left tilt. The above results suggest that the stepping foot is fixed at MER and BR, and pitching motion is performed by pelvic movements around the hip joint. Based on the results of the orientation and the anterior/lateral tilt of the pelvis and thorax, it is considered that female pitchers’ pelvis and thorax perform horizontal rotational movements in the posture horizontal to the ground. On the other hand, male pitchers’ pelvis and thorax perform oblique rotational movements in the anterior and left tilting posture (Figure 6).

When performing oblique rotational movements with the pelvis and thorax anterior/left tilt, the center of mass of the upper body moves to the anterolateral part, compared to the posture without the tilt. Therefore, in the follow-through phase, the gluteus maximus muscle and hamstrings of the lead leg as well as the lower trapezius muscle that act upon the pelvis, thorax, and the upper body are used much more than horizontal rotational movements. Generally, women are supposed to have less muscle strength than men, and it is considered difficult for women to perform a pitching motion with oblique rotational movements, with the pelvis and thorax anterior/left tilting. When coaching pitching motion for female players, it would be useful to evaluate the muscle strength of the lower limbs and the trunk and use horizontal rotation and oblique rotational movements depending on the muscle strength.

### 4.4. Transmitting the Energy

One limitation of this study was that correlations between physical functions and pitching motion were not examined. Movements of the pelvis and thorax are affected not only by the lead leg and trunk functions but also by translational motion performed before rotational movements that appear after foot contact. The kinetic chain of the lower extremity, trunk, and upper extremity is considered important in pitching motions [2]. Moreover, the stability of the pivoting foot is important for transmitting the energy produced by translational motion during the early cocking phase to the lead leg, the trunk, and the throwing-side of the upper extremity [17]. Therefore, it would be useful to examine correlations between flexibility/muscle strength of the upper/lower extremity as well as the trunk and horizontal/oblique rotational movements when coaching pitching motion. There is also a need to investigate correlations between above factors and performance indices such as the speed of the ball when pitching.

## 5. Conclusions

Pitching motion of professional female baseball pitchers that could throw a ball with their full strength and those of male university pitchers were compared. The pelvis and thorax of female pitchers were significantly directed to the first base at both MER and BR, compared to male pitchers. Moreover, lateral tilt angles of the pelvis and thorax of female pitchers were lower than male pitchers at both MER and BR. The results of this study indicate that the pelvic and thoracic movements of the professional female baseball pitchers was different from male university pitchers.

## Figures and Tables

**Figure 1 ijerph-18-13342-f001:**
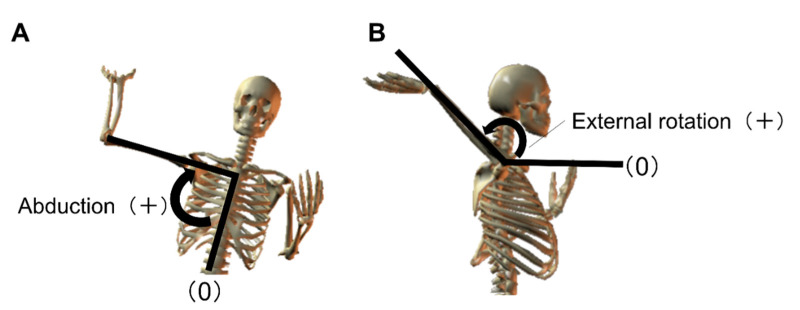
Shoulder joint angle. (**A**): abduction, (**B**): external rotation.

**Figure 2 ijerph-18-13342-f002:**
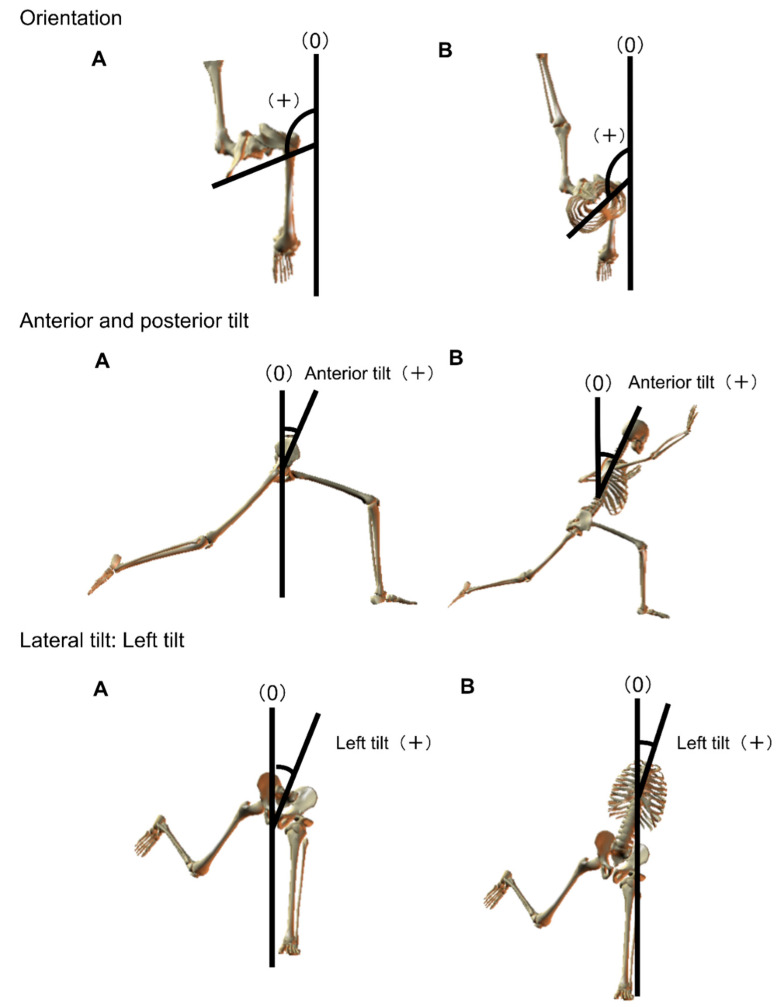
Orientation and tilt of the pelvis and thorax. (**A**): pelvis, (**B**): thorax.

**Figure 3 ijerph-18-13342-f003:**
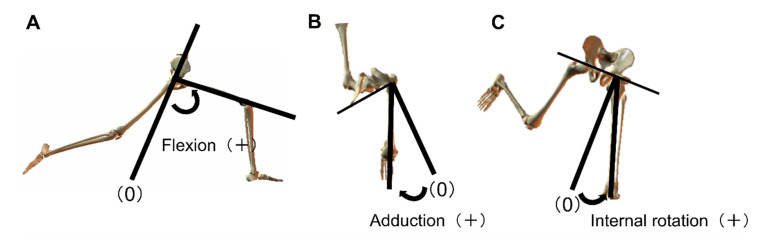
Hip joint angle. (**A**): flexion, (**B**): adduction, (**C**): internal rotation.

**Figure 4 ijerph-18-13342-f004:**
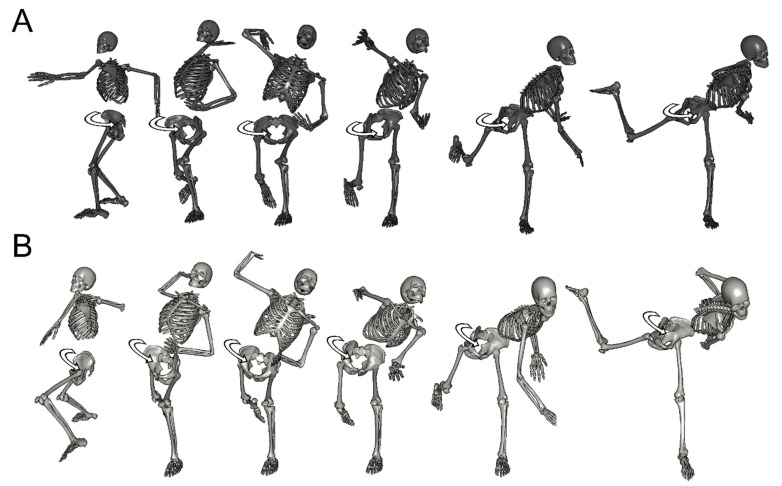
The pitching form of the representative example watched from home base. (**A**): Women/horizontal rotational movement, (**B**): men/oblique rotational movement.

**Figure 5 ijerph-18-13342-f005:**
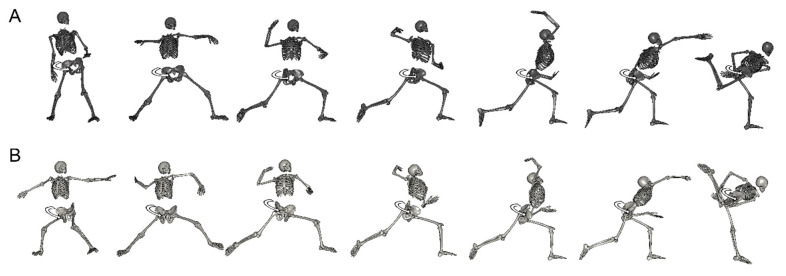
The pitching form of the representative example watched from third base (**A**): women/horizontal rotational movement, (**B**): men/oblique rotational movement.

**Figure 6 ijerph-18-13342-f006:**
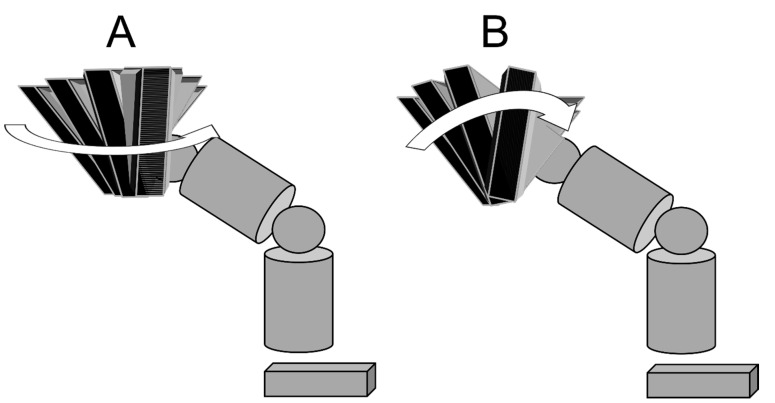
Horizontal rotational movement and oblique rotational movement of the pelvis. (**A**): Women/horizontal rotational movement, (**B**): men/oblique rotational movement.

**Table 1 ijerph-18-13342-t001:** Attributes of the participants.

	Female Professional Baseball Pitchers (Mean ± SD, N = 15)	Male University Baseball Pitchers (Mean ± SD, N = 14)	*p*-Value
Age (years)	21.7 ± 3.2	19.9 ± 0.8	0.06
Height (cm)	162.5 ± 5.1	176.4 ± 3.0	0.01 *
Body mass (Kg)	59.0 ± 6.6	73.1 ± 3.0	0.00 *
Years of baseball experience	14.5 ± 3.2	11.6 ± 2.1	0.01 *
Years of experience as a pitcher	12.4 ± 3.5	8.6 ± 3.1	0.01 *

*: Significant differences, *p* < 0.05.

**Table 2 ijerph-18-13342-t002:** Shoulder joint angle.

	Female Professional Baseball Pitchers		Male University Baseball Pitchers	
	MER	BR	MER	BR
Abduction (°)	94.8 ± 8.3	92.5 ± 8.5	100.1 ± 11.0	96.4 ± 8.9
External rotation (°)	161.8 ± 12.4	129.8 ± 13.3	166.0 ± 13.5	132.9 ± 12.2

Mean ± SD. MER, maximum external rotation; BR, ball release.

**Table 3 ijerph-18-13342-t003:** Orientation of the pelvis and thorax.

	Female Professional Baseball Pitchers		Male University Baseball Pitchers	
	MER	BR	MER	BR
Orientation				
Pelvis (°)	93.7 ± 4.7	99.5 ± 5.4	85.4 ± 5.4	92.4 ± 5.8
Thorax (°)	102.9 ± 6.7	114.4 ± 6.3	93.7 ± 6.3	104.9 ± 7.3
Thorax-pelvis (°)	9.3 ± 5.9	14.8 ± 5.8	8.3 ± 7.1	12.5 ± 6.8

Mean ± SD. MER, maximum external rotation; BR, ball release.

**Table 4 ijerph-18-13342-t004:** Tilt of the pelvis and thorax.

	Female Professional Baseball Pitchers		Male University Baseball Pitchers	
	MER	BR	MER	BR
Anterior and posterior tilt				
Pelvis (°)	35.6 ± 5.5	36.7 ± 5.9	37.4 ± 14.2	37.4 ± 14.7
Thorax (°)	17.0 ± 5.1	26.5 ± 5.1	24.0 ± 9.2	33.1 ± 8.6
Thorax-pelvis (°)	−18.7 ± 8.2	−10.2 ± 7.9	−13.3 ± 16.6	−4.3 ± 17.2
Lateral tilt				
Pelvis (°)	1.2 ± 5.2	1.6 ± 4.6	8.8 ± 6.0	9.6 ± 6.7
Thorax (°)	24.4 ± 6.5	28.5 ± 7.9	30.4 ± 6.5	33.3 ± 6.6
Thorax-pelvis (°)	23.2 ± 8.9	26.9 ± 10.3	21.5 ± 7.7	23.8 ± 8.1

Mean ± SD. MER, maximum external rotation; BR, ball release.

**Table 5 ijerph-18-13342-t005:** Hip joint angle.

	Female Professional Baseball Pitchers		Male University Baseball Pitchers	
	MER	BR	MER	BR
Flexion (°)	99.4 ± 10.3	98.2 ± 10.5	105.3 ± 18.1	101.4 ± 19.0
Adduction (°)	7.2 ± 6.9	10.5 ± 7.1	−4.7 ± 5.1	−0.6 ± 4.6
Internal rotation (°)	19.1 ± 11.4	17.5 ± 12.0	32.7 ± 11.5	31.6 ± 13.2

Mean ± SD. MER, maximum external rotation; BR, ball release.

## Data Availability

The datasets used and/or analyzed during the current study are available from the corresponding author on reasonable request.

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
