# Peer review of "Motion Analysis Focusing on Rotational Movements of Professional Female Baseball Pitchers: Comparison with Male University Baseball Pitchers"

_ijerph, 2021, doi:10.3390/ijerph182413342_

Round 1

Reviewer 1 Report

Dear Authors:

Major revisions are proposed

King Regards

Author Response

Thank you for your letter. We are grateful for the detailed feedback provided by the reviewers, which we feel has helped us to significantly improve the paper. Attached are our point-by-point responses to the comments from the reviewers and our revised manuscript, which we hope will now meet with your approval. For your convenience, we have attached a copy of the manuscript with all revisions highlighted in red font. We believe that our revisions have addressed the issues raised by the reviewers and trust that the manuscript will now prove suitable for publication in the International Journal of Environmental Research and Public Health.

Reviewer 2 Report

I enjoyed reading your paper on baseball and the comparison between males and females.

Below I have made some specific comments and recommendations.

Abstract:

It would be helpful to show some data in the abstract, not just the p-values.

The conclusion isn’t overly clear. Are you making a recommendation or is it an observation in line 33/34?

Line 34: Baseball pitchers is repeated, please delete.

Introduction:

You provide a succinct introduction and explain pitching. I would like to see some additional information about the type and frequency of injuries (men vs women), severity and time away from the game (if available). Also, so information about the different throwing speeds between genders would be helpful, as well as the total number of players.  

Can you expand your aims by including what you anticipated you would find?

Methods:

The methods are described sufficiently, and the illustrations are helpful. If possible, pictures of the actual set-up and an athlete throwing should be added.

Results:

You show the results clearly in text and tables. Using a model, would it be possible to illustrate the different angles in the genders for one throw to allow for better comparison? I mean two models, one male and one female, at the different stages of a throw. I think this would nicely highlight the differences. You make reference to the differences in figure 4, but using a whole body model this might be even better explained.

I’m aware that ball speed was not a major concern of yours, but are there any possible correlations to be calculated from these?

And as you are interested in injuries, did you collect any data from the players about this? Again, some valuable correlations could perhaps be calculated.

Discussion:

Line 17 add full stop after et al.

Line 20. Please provide a reference for this statement

The conclusion needs to be more informative. Currently there are some suggestions, but the practical application of the work should be further highlighted.

You mention injuries and differences in movement in the introduction, but do not address this again. I feel this is something that should be addressed, even if just briefly.

Author Response

(The authors gave the same response as above.)

Round 2

Reviewer 1 Report

Accepted